# Identification of a High-Risk Group of New-Onset Cardiovascular Disease in Occupational Drivers by Analyzing Heart Rate Variability

**DOI:** 10.3390/ijerph182111486

**Published:** 2021-10-31

**Authors:** Ying-Chuan Wang, Chung-Ching Wang, Ya-Hsin Yao, Wei-Te Wu

**Affiliations:** 1Division of Family Medicine, Department of Family and Community Medicine, Tri-Service General Hospital, National Defense Medical Center, Taipei 114, Taiwan; popoga96@gmail.com (Y.-C.W.); bigching@gmail.com (C.-C.W.); 2Division of Occupational Medicine, Department of Family and Community Medicine, Tri-Service General Hospital, National Defense Medical Center, Taipei 114, Taiwan; 3School of Medicine, National Defense Medical Center, Taipei 114, Taiwan; yamcom13579@gmail.com; 4National Institute of Environmental Health Sciences, National Health Research Institutes, Miaoli 350, Taiwan; 5Institute of Environmental and Occupational Health Sciences, National Yang Ming Chiao Tung University, Taipei 112, Taiwan; 6Department of Healthcare Administration, Asia University, Taichung 413, Taiwan

**Keywords:** early monitoring, heart rate variability, cardiovascular diseases, professional bus driver, cohort study

## Abstract

Purpose: This cohort study evaluated the effectiveness of noninvasive heart rate variability (HRV) analysis to assess the risk of cardiovascular disease over a period of 8 years. Methods: Personal and working characteristics were collected before biochemistry examinations and 5 min HRV tests from the Taiwan Bus Driver Cohort Study (TBDCS) in 2005. This study eventually identified 161 drivers with cardiovascular disease (CVD) and 627 without between 2005 and 2012. Estimation of the hazard ratio was analyzed by using the Cox proportional-hazards model. Results: Subjects with CVD had an overall lower standard deviation of NN intervals (SDNN) than their counterparts did. The SDNN index had a strong association with CVD, even after adjusting for risk factors. Using a median split for SDNN, the hazard ratio of CVD was 1.83 (95% CI = 1.10–3.04) in Model 1 and 1.87 (95% CI = 1.11–3.13) in Model 2. Furthermore, the low-frequency (LF) index was associated with a risk of CVD in the continuous approach. For hypertensive disease, the SDNN index was associated with increased risks in both the continuous and dichotomized approaches. When the root-mean-square of the successive differences (RMSSDs), high frequency (HF), and LF were continuous variables, significant associations with hypertensive disease were observed. Conclusions: This cohort study suggests that SDNN and LF levels are useful for predicting 8 year CVD risk, especially for hypertensive disease. Further research is required to determine preventive measures for modifying HRV dysfunction, as well as to investigate whether these interventions could decrease CVD risk among professional drivers.

## 1. Introduction

Cardiovascular disease (CVD) is not only the leading cause of death in the world, but also a compensable diseases related to work [1,2]. Since the 1950s, many researchers have studied the occupational issues of professional drivers [3]. Male professional drivers have an elevated morbidity and mortality from myocardial infarction (MI), ischemic heart disease (IHD), coronary heart disease (CHD) [4,5,6], stroke [7], and arteriosclerosis, according to the brachial–ankle pulse wave velocity [8]. Some studies have indicated that professional drivers have an elevated risk of developing CVD due to a high workload and a psychosocial work environment, due to a highly demanding job, irregular shifts, overtime work, and limited meals and rest time [6,9,10,11]. Professional drivers living with CVD conditions are predisposed to work stress, triggering death by overwork. Therefore, early monitoring of a high-risk CVD group is important to design preventive measures and thus limit further health damage in the workplace.

The examination of heart rate variability (HRV) is a simple, noninvasive, and relatively inexpensive method for an epidemiological study with a large sample size [12,13,14,15,16,17,18]. HRV measures specifically reflect vagal activity and have been recommended by the Task Force of the European Society of Cardiology and the North American Society of Pacing and Electrophysiology (1996) [19]. The cardiovascular system is controlled by the nervous system, specifically, the autonomic nervous system (ANS) [18,19,20,21]. Appendix A (available as an online Appendix A) presents the definitions of HRV measures applied in our research [22,23].

Reduced HRV as a marker of autonomic dysfunction has been shown to be associated with a poor prognosis of CVD, as well as with MI incidence, CVD mortality, and death from other causes in the general population [24,25,26,27,28,29,30,31], Furthermore, decreased HRV at rest is associated with a poor prognosis of CVD [32], and reduced resting HRV is considered a risk marker for future cardiovascular and other stress-related diseases [33]. 

However, some problems have emerged in this research field, including small sample sizes, incomplete CVD data collection, and poor control for confounders, which have limited the evaluation of the independent predictive effect of HRV, and has not shown a clear causal relationship. Therefore, we adopted a cohort study design to assess the effectiveness of noninvasive HRV analysis to measure professional drivers’ autonomic function, and then investigated the relationship between HRV and the 8 year risk of CVDs.

## 2. Materials/Subjects and Methods

### 2.1. Study Population

A Taiwan Bus Drivers Cohort (TBDC) has previously been established [34] for a longitudinal follow-up study. We linked this cohort to Taiwan’s National Health Insurance Research Database (NHIRD) to obtain the medical information of these subjects. This study was approved by the Institutional Review Board of the National Health Research Institutes, Taiwan (NIRB File Number: EC1060516-E). The composition and operation of the review committee were established in accordance with the International Conference on Harmonization–Good Clinical Practice (ICH-GCP) guidelines. The authors confirm that all experiments were carried out in accordance with the relevant guidelines and regulations. Informed consent was obtained from all participants. A questionnaire was used to collect basic information and working patterns, including demographic characteristics, work conditions (year of first employment and bus driving experience), lifestyle habits, and job stress assessment. This study adopted a longitudinal design from 2005 to 2012, the questionnaire collected information on basic and working patterns and HRV, and two sets of biochemical measurements were conducted simultaneously (Appendix A). This cohort was linked to Taiwan’s National Health Insurance Research Database (NHIRD) to obtain the CVD medical information of these subjects.

Figure 1 illustrates the procedures used in this study. The TBDC was created in 2006 and includes 1650 professional drivers from the largest transportation company in Taiwan. The Driving Hours Dataset between 2005 and 2007 was used to exclude subjects with a driving duration of fewer than 100 days (*n* = 613). Then, personal and working characteristics were collected before biochemistry examinations, and HRV tests were performed from 2007 to 2008. We excluded individuals with incomplete questionnaires or laboratory data (*n* = 249). Subsequently, we linked the remaining 788 drivers to the ambulatory care expenditures-by-visits and inpatient expenditures-by-admissions data from the NHIRD. The defining criteria for CVD cases were that bus drivers had had at least five recorded clinical visits within one year due to CVD, or at least one inpatient record because of CVD for the first-listed diagnosis code. The strict criteria increased the sensitivity and decreased the specificity for confirming CVD. We identified 161 drivers with CVD (International Classification of Diseases 9th Revision, ICD-9: 390–459) and 627 drivers without CVD from 2005 to 2012. Among the 161 drivers with CVD, 84 had a history of CVD before 2006. Finally, 77 incident CVD cases were defined. Meanwhile, CVD (excluding hypertensive disease) (ICD-9: 391, 392.0, 393–398, 410–414, 416, 420–429), IHD (ICD-9: 410–414), cerebrovascular disease (ICD-9: 430–438), and congestive heart failure (CHF) (ICD-9: 398.91, 422, 425, 428, 402.x1, 404.x1, 404.x3) were analyzed separately. 

### 2.2. HRV and Biochemical Measurements

Each participant underwent a blood biochemistry test and noninvasive HRV examination in resting conditions with the ANS Analyzer (Medicore SA-3000P, Jamsil-dong, Songpa-gu, Seoul, Korea). The variability in heart rate over 5 min was analyzed by the method of time domain and frequency domain. This provided the degree of balance and activity of the ANS. The standard deviation of the normal-to-normal beats interval (SDNN) and the square root of the mean squared differences of successive N–N intervals (RMSSD) were used to compare the time domain indexes. Frequency domain methods, including very low frequency (VLF, 0.0033–0.04 Hz), low frequency (LF, 0.04–0.15 Hz), high frequency (HF, 0.15–0.4 Hz), and total power (TP) were used to determine the sympathetic and parasympathetic heartbeat rate modulations at rest. The physical stress index (PSI) reflected the load and pressure on the heart based on SDNN at the same time [35].

For the determination of total cholesterol (CHOL), this study employed the cholesterol oxidase method on an AU640 analyzer (Beckman Coulter Ltd., High Wycombe, UK). High-density lipoprotein cholesterol (HDL-C) levels were determined using the immunoinhibition method on the AU640 analyzer (Beckman Coulter Ltd., High Wycombe, UK). Triglyceride (TG) concentrations were determined using an enzymatic method on the AU640 analyzer (Beckman Coulter Ltd., High Wycombe, UK). Fasting blood glucose (FG) was conducted using the hexokinase method, also on the AU640 analyzer (Beckman Coulter Ltd., High Wycombe, UK).

### 2.3. Statistical Analysis

Logarithmical transformation was performed to approximate the normal distribution. This study also used a Cox proportional-hazards model to assess the effect of HRV parameters on the risk of CVD (hazard ratios (HRs) and 95% confidence intervals (CIs)) and to adjust for confounding variables. Standard median splits were used on HRV parameters (the continuous variables) to turn them into dichotomous variables. The risk factors we considered included age, job tenure, shift work, body mass index, drinking, smoking, exercise, and education. Moreover, clinical conditions such as systolic blood pressure, CHOL, TG, HDL, and fasting glucose were also considered. Age at first employment (≥45 vs. <45 years), time since first employment (years), shift work, body mass index (BMI; >30 vs. ≤30), drinking, smoking, exercise, and education were adjusted in Model 1. Next, we adjusted for clinical conditions, including systolic blood pressure, LnCHOL, LnTG, LnHDL, and Ln (fasting glucose) in Model 2. All analyses were performed using the Statistical Analysis System (SAS) software package (Version 9.3 for Windows; SAS Institute Inc., Cary, NC, USA).

## 3. Results

Demographic characteristics of the bus drivers are presented in Table 1. A total of 788 drivers and 5334.2 person-years were accumulated in this cohort. Almost half of the bus drivers were over 40 years old in their first employment (43.3%), more than half of the bus drivers (51.5%) had over 5 years of driving experience, and almost half were irregular shift-working drivers (47%). About 16% of the bus drivers were obese (BMI ≥ 30 kg/m^2^), more than half of the bus drivers had a smoking habit (57.5%), and 21.7% of the bus drivers had a drinking habit.

A comparison of HRV parameters between different cardiovascular diagnostic categories is shown in Appendix A (available as an online Appendix A). The cohort of 788 subjects included 49 people with CVD (not including hypertensive disease); 128 people with hypertensive disease; 35 people with IHD; 14 people with cerebrovascular disease; 8 people with diseases of the arteries, arterioles, and capillaries, as well as other diseases of the circulatory system; and 15 people with CHF.

### 3.1. HRV Indices and 8-Year CVD Risks

Table 2 lists the hazard ratios for CVD per single unit increment of HRV parameters (as continuous variables), as well as for dichotomized HRV parameters. For the 788 drivers with a known CVD history, an increased SDNN level had a negative association with the risk of CVD in the continuous approach in both models. The SDNN had a significant hazard ratio (per single unit increment) of 0.67 to 0.70. Regarding the dichotomized approach with a median split, a low SDNN level was associated with CVD (hazard ratio = 1.47; 95% CI = 1.04–2.07) in Model 1 and (1.44; 95% CI = 1.01–2.05) in Model 2. 

Similar to the aforementioned findings, among the 704 drivers without a known CVD history at baseline, the SDNN index continued to exhibit a statistically significant association with the risk of CVD. In Model 2, a single unit increment in Ln SDNN was associated with a decrease of 44% in the hazard for CVD, with adjustments for demographics, working characteristics, and clinical risk factors (95% CI = 0.34–0.95, *p* = 0.031). Regarding the dichotomized approach with a median split, a low SDNN was associated with a hazard ratio of 1.83 (95% CI = 1.10–3.04) in Model 1 and 1.87 (95% CI = 1.11–3.13) in Model 2. Furthermore, the LF index exhibited associations with the risk of CVD in the continuous approach in both models.

### 3.2. HRV Indices and 8 Year Cardiovascular Diagnostic Risk Categories

Table 3 and Appendix A list the hazard ratios of HRV indices for cardiovascular diagnostic categories among the different driver groups with or without a known CVD history at baseline. After we excluded 84 cases of prevalent CVD before 2006 (Table 3), we found that the SDNN index was associated with increased risks of hypertensive disease in both the continuous and dichotomized approaches. A single unit increment in Ln SDNN was associated with a decrease of 65% in hypertensive disease in both models (Model 1: 95% CI = 0.19–0.66, *p* = 0.001; Model 2: 95% CI = 0.19–0.67; *p* = 0.002). Low levels of SDNN (0–30) were associated with increased risks of hypertensive disease in both models (Model 1: hazard ratio = 1.99; 95% CI = 1.03–3.84; *p* = 0.039; Model 2: hazard ratio = 2.02; 95% CI = 1.03–3.96; *p* = 0.041). 

Meanwhile, a single unit increment in Ln RMSSD was associated with a decrease of 45–46% in hypertensive disease in the two models (Model 1: hazard ratio = 0.54; 95% CI = 0.31–0.92, *p* = 0.024; Model 2: hazard ratio = 0.55; 95% CI = 0.31–0.96; *p* = 0.035). 

A single unit increment in Ln HF was associated with a decrease of 26–27% in hypertensive disease in two models (Model 1: hazard ratio = 0.73; 95% CI = 0.57–0.94, *p* = 0.015; Model 2: hazard ratio = 0.74; 95% CI = 0.57–0.96; *p* = 0.026). Ln LF had a significant hazard ratio of 0.76 for hypertensive disease in Model 1 (95% CI = 0.59–0.97; *p* = 0.027), which became nonsignificant in Model 2.

For CHF, Ln RMSSD only had a significant hazard ratio of 3.51, for which in Model 2: 95% CI = 1.03–12.0; *p* = 0.046.

## 4. Discussion

This study used a prospective professional cohort study to investigate the relationship between HRV and the risk of CVD in professional drivers without known CVD. The major finding of this study was that the SDNN and LF levels are useful for predicting the 8 year CVD risk even when adjusting for CVD risk factors. Furthermore, increased SDNN and LF levels elevated the risks for other CVD events, such as hypertensive disease. 

Each unit increment in Ln SDNN was associated with a decrease of 65% in hypertensive disease in Model 2 (95% CI = 0.19–0.67, *p* = 0.002). Our results are consistent with a meta-analysis that indicated that the predicted risks of incident CVD of the 10th and 19th percentiles in SDNN compared with the 50th percentile were 1.50 (95% CI = 1.22, 1.83) and 0.67 (95% CI = 0.41, 1.09), respectively [36]. In general, the SDNN is the gold standard for the medical stratification of cardiac risk and predicts both CVD morbidity and mortality. However, this only applies when recorded over a 24 h period. 

Furthermore, this study observed that the LF index was associated with the risk of CVD and hypertensive disease in a continuous approach. While sitting upright during resting conditions, the LF reflected parasympathetic nervous system activity and baroreflex activity, not sympathetic nervous system activity and cardiac sympathetic innervation [22]. A previous study demonstrated that a higher occupational workload resulted in reduced LF power, which indicates that a high workload is related to attenuating cardiac autonomic modulation during sleep. In contrast, enhanced sympathetic baroreceptor cardiac regulation during sleep in workers with a high level of physical leisure time activity was observed [37]. Bus drivers have a high workload and less leisure-time activity, which leads to the development of CVDs; thus, low LF power is reflected in advance.

Additionally, drivers with a low HRV may already suffer from silent CVD. Numerous overlapping risk factors exist for reduced HRV and CVD events [38]. However, the causal relationship of risk factors with the development of CVD or reduced HRV is still not completely understood. Work stress is associated with both CVD and reduced HRV [39]; however, we do not yet know whether work stress affects the development of CVD more than it contributes to reduced HRV. Further investigating the associations between psychosocial risk factors and HRV indices would be worthwhile [40,41]. Psychosocial conditions such as work stress, stressful life events, and mood disorders are emerging risk factors for CVD [42]. Risk factors are preceded by indicators of decreased vagal function; therefore, HRV was found to be a useful tool for studying work-related stress and the accompanying physiological effects. The SDNN was reported to be significantly lower among those categorized into a high-job-strain group than among those categorized into a low-job-strain group [43]. Amelsvoort [39] reported that a decreased SDNN level in shift workers indicates less favorable cardiovascular autonomic regulation. Moreover, numerous studies have indicated that a chronic autonomic imbalance with sympathetic dominance may partially explain the effects of work stress on CVD events [21]. Therefore, HRV could be used to screen workers at high risk of CVD, and preventive measures could be taken in advance. 

This study had several advantages that included the large bus drivers cohort, the prospective design, the noninvasive marker of 5 min HRV measurements, confounders’ adjustment, and comprehensive CVD data collection. In addition, we fully admit that the method of this study has some limitations. First, only male bus drivers were included, which restricts the generalization of the results to females. Second, HRV may be influenced by the severity of CVD, respiratory patterns, circadian rhythm, as well as by the use of β-blockers or antidepressants [44,45,46,47]. Thus, analyses should be further stratified by the severity of diseases, such as MI or revascularization, as well as by ICD-10-PCS (Procedure Codes). In addition, a history of diabetes, cognitive disorders, severe lung diseases, and the use of β-blockers and antidepressants must be considered. Third, the current study design could not clarify which risk factors contributed more to reduced HRV and CVD events so that preventive measures can be taken in advance. The small number of subjects in these sub-categories is a restriction of this study. The statistical power may be too small to generalize the results to other subjects. Fourth, there is a possibility that the high false positive rate in this study could have caused incorrect results. Therefore, we used the strict criteria that CVD cases had to have at least five visits for the same diagnosis within 1 year, or were an inpatient with one or more admissions during the study period based on a clinical physician’s suggestion. This increased the sensitivity and decreased the specificity for CVD, but it could have underestimated the effect of our final result. Lastly, and most importantly, this study directly used commercial instruments and automated programs to analyze the change in heart rate during the short term by the method of time domain and frequency domain with an ANS Analyzer. This may obscure some meaningful signals in heart rate for arrhythmias or other CVD diseases. With the technological advances in big data analytics, a future study should be attempted to identify and interpret the computer identification of segments with aberrant patterns for CVD diseases.

In conclusion, this professional drivers’ cohort study concluded that the HRV parameters SDNN and LF are independent predictors of CVD and hypertensive disease, even after adjusting for risk factors. Further research is required to determine the preventive measures for modifying HRV dysfunction, as well as to investigate whether these interventions could reduce CVD risk in professional drivers.

## Figures and Tables

**Figure 1 ijerph-18-11486-f001:**
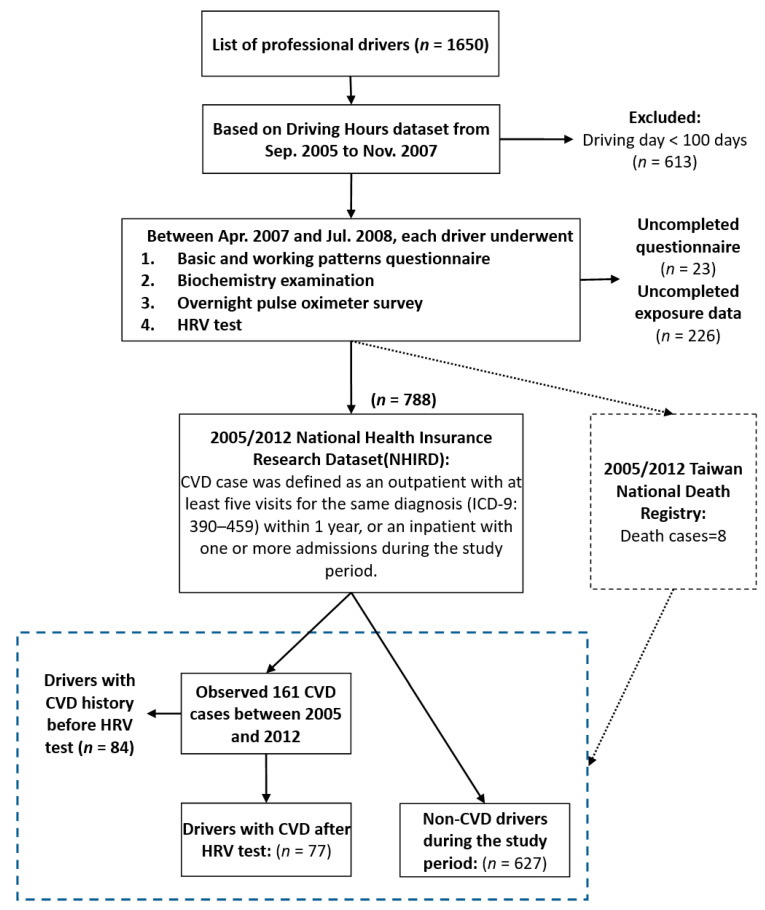
Study flow diagram in the Taiwan Bus Driver Cohort Study.

**Table 1 ijerph-18-11486-t001:** Baseline characteristics of the study population.

Variables	All Drivers		Person Years	
	*n*	(%)	sum	(%)
Total subjects	788	100.0	5334.2	100.0
Non-CVD drivers	627	79.6	5014.3	94.0
CVD drivers ^a^	161	20.4	319.9	6.0
CVD history before 2006 ^a,b^	84	10.7	11.7	0.2
Age (years)				
<35	87	11.0	666.5	12.5
35–44	340	43.1	2417.4	45.3
45–49	199	25.3	1339.6	25.1
≥50	162	20.6	910.7	17.1
Age at first employment (years)				
≤32	175	22.2	1320.6	24.8
33–38	272	34.5	1872.6	35.1
≥39	341	43.3	2141.1	40.1
Time since first employment (years)				
≤2	150	19.0	1091.8	20.5
2.1–5	232	29.4	1647.2	30.9
5.1–8	164	20.8	1059.9	19.9
>8	242	30.7	1535.4	28.8
Shift work modes ^c^				
Day shifts only	338	42.9	2264.8	42.5
Irregular shift	370	47.0	2587.1	48.5
Evening and Night shift	80	10.2	482.4	9.0
BMI (kg/m^2^)				
<25	299	37.9	2166.6	40.6
25–29.9	359	45.6	2361.8	44.3
≥30	130	16.5	805.9	15.1
Marital status				
Unmarried	124	15.7	919.8	17.2
Married	577	73.2	3841.7	72.0
Others	87	11.0	572.7	10.7
Education				
≤Junior high school	235	29.8	1556.9	29.2
Senior high and vocational school	498	63.2	3396.2	63.7
University and College	55	7.0	381.1	7.1
Cigarette smoking				
Current smokers	276	35.0	1808.7	33.9
Ex-smokers	54	6.9	337.4	6.3
Never smokers	453	57.5	3148.1	59.0
Missing	5			
Alcohol use				
Yes	612	77.7	4240.5	79.5
No	171	21.7	1061.3	19.9
Missing	5			
Moderate exercise				
Yes	557	70.7	3857.0	72.3
No	221	28.0	1397.3	26.2
Missing	10			

^a^ The selection criteria for CVD (ICD-9-CM: 390–459) were at least five clinical visit records within a year or at least one inpatient record; ^b^ drivers who had a CVD history before 2006; ^c^ based on the Driving Hours Dataset.

**Table 2 ijerph-18-11486-t002:** Hazard ratios and 95% confidence intervals for cardiovascular disease by HRV index in the study population.

		All Drivers (*n* =788)								Drivers (*n* = 704) ^a^							
		Model 1 ^b^				Model 2 ^c^				Model 1 ^b^				Model 2 ^c^			
	Independent Variables ^d^	HR	95% CI		*p*-Value	HR	95% CI		*p*-Value	HR	95% CI		*p*-Value	HR	95% CI		*p*-Value
1	As a continuous (LnSDNN)	0.67	0.48	0.93	0.018	0.70	0.50	1.00	0.047	0.57	0.35	0.95	0.029	0.56	0.34	0.95	0.031
2	As a categorical variable: SDNN (≤30 vs. >30)	1.47	1.04	2.07	0.029	1.44	1.01	2.05	0.044	1.83	1.10	3.04	0.020	1.87	1.11	3.13	0.018
3	As a continuous (LnRMSSD)	0.83	0.62	1.10	0.185	0.85	0.64	1.13	0.264	0.83	0.54	1.28	0.397	0.81	0.52	1.26	0.348
4	As a categorical variable: RMSSD (≤20 vs. >20)	1.34	0.95	1.89	0.098	1.34	0.94	1.91	0.104	1.34	0.81	2.20	0.256	1.38	0.83	2.28	0.211
5	As a continuous (LnLF)	0.85	0.74	0.97	0.016	0.88	0.76	1.02	0.084	0.80	0.66	0.98	0.031	0.79	0.64	0.98	0.032
6	As a categorical variable: LF (≤380 vs. >380)	1.18	0.79	1.74	0.420	1.14	0.76	1.69	0.535	1.25	0.70	2.23	0.445	1.25	0.70	2.25	0.453
7	As a continuous (LnHF)	0.91	0.79	1.04	0.176	0.93	0.81	1.06	0.283	0.84	0.69	1.03	0.098	0.84	0.68	1.04	0.112
8	As a categorical variable: HF (≤168 vs. >168)	1.05	0.72	1.54	0.786	1.07	0.72	1.58	0.743	0.98	0.58	1.67	0.949	1.00	0.58	1.71	0.996
9	As a continuous (LnLF/HF)	0.90	0.76	1.06	0.212	0.94	0.80	1.11	0.486	0.93	0.73	1.18	0.541	0.93	0.72	1.19	0.544
10	As a categorical variable: LF/HF (≤3.5 vs. >3.5)	1.27	0.90	1.78	0.173	1.16	0.82	1.63	0.409	1.24	0.75	2.03	0.405	1.20	0.72	1.97	0.486

^a^ Excluded 84 drivers who had a CVD history from before 2006; ^b^ Model 1: Adjusted for age at first employment (≥45 vs. <45 years), body mass index (>30 vs. ≤30), education, drinking, smoking, exercise, time since first employment (years), and shift work; ^c^ Model 2: Same as Model 1, with additional adjustments for systolic blood pressure, LnCHOL, LnTG, LnHDL, and Ln (fasting sugar); ^d^ each independent variable (1–20) was separately included in the models.

**Table 3 ijerph-18-11486-t003:** Hazard ratios and 95% confidence intervals for cardiovascular events by HRV index in the Scheme 704 ^a^.

			Cardiovascular Disease (Not Including Hypertensive Disease)				Hypertensive Disease				Ischemic Heart Disease				Congestive Heart Failure (CHF)			
		Independent Variables ^d^	HR	95% CI		*p*-Value	HR	95% CI		*p*-Value	HR	95% CI		*p*-Value	HR	95% CI		*p*-Value
Model 1 ^b^	1	As a continuous (LnSDNN)	1.44	0.59	3.55	0.423	0.35	0.19	0.66	0.001	1.12	0.35	3.54	0.851	2.46	0.64	9.42	0.188
	2	As a categorical variable: SDNN (≤30 vs. >30)	1.61	0.64	4.05	0.316	1.99	1.03	3.84	0.039	1.36	0.45	4.14	0.584	1.96	0.35	10.95	0.441
	3	As a continuous (LnRMSSD)	2.06	1.01	4.21	0.048	0.54	0.31	0.92	0.024	2.02	0.81	5.03	0.133	2.92	0.92	9.29	0.069
	4	As a categorical variable: RMSSD (≤20 vs. >20)	0.80	0.34	1.91	0.615	1.87	0.94	3.70	0.074	0.64	0.23	1.79	0.392	0.82	0.18	3.71	0.795
	5	As a continuous (LnLF)	1.05	0.73	1.51	0.783	0.76	0.59	0.97	0.027	0.96	0.61	1.50	0.855	1.30	0.64	2.66	0.470
	6	As a categorical variable: LF (≤380 vs. >380)	1.01	0.36	2.80	0.988	1.31	0.62	2.74	0.479	1.13	0.31	4.11	0.852	0.42	0.07	2.73	0.366
	7	As a continuous (LnHF)	0.99	0.69	1.42	0.971	0.73	0.57	0.94	0.015	1.15	0.73	1.80	0.543	0.90	0.45	1.80	0.764
	8	As a categorical variable: HF (≤168 vs. >168)	0.71	0.28	1.77	0.460	1.29	0.63	2.64	0.492	0.66	0.22	1.98	0.453	0.54	0.09	3.26	0.502
	9	As a continuous (LnLF/HF)	1.09	0.71	1.67	0.708	1.04	0.77	1.42	0.786	0.78	0.47	1.32	0.358	1.70	0.72	4.00	0.228
	10	As a categorical variable: LF/HF (≤3.5 vs. >3.5)	0.95	0.40	2.26	0.901	1.02	0.56	1.85	0.954	1.34	0.42	4.28	0.621	0.77	0.17	3.53	0.740
Model 2 ^c^	11	As a continuous (LnSDNN)	1.70	0.64	4.52	0.290	0.35	0.19	0.67	0.002	1.04	0.30	3.66	0.947	3.18	0.75	13.47	0.117
	12	As a categorical variable: SDNN (≤30 vs. >30)	1.61	0.62	4.18	0.332	2.02	1.03	3.96	0.041	1.40	0.44	4.41	0.568	1.94	0.30	12.57	0.485
	13	As a continuous (LnRMSSD)	2.17	1.03	4.59	0.043	0.55	0.31	0.96	0.035	2.30	0.88	5.98	0.089	3.51	1.03	12.02	0.046
	14	As a categorical variable: RMSSD (≤20 vs. >20)	0.79	0.33	1.91	0.600	1.92	0.96	3.87	0.067	0.57	0.20	1.68	0.310	0.55	0.09	3.42	0.519
	15	As a continuous (LnLF)	1.02	0.68	1.51	0.936	0.77	0.59	1.01	0.057	0.90	0.55	1.47	0.674	1.35	0.59	3.10	0.477
	16	As a categorical variable: LF (≤380 vs. >380)	1.03	0.36	2.91	0.960	1.21	0.57	2.58	0.620	1.19	0.32	4.41	0.790	0.29	0.03	2.67	0.272
	17	As a continuous (LnHF)	1.03	0.70	1.53	0.880	0.74	0.57	0.96	0.026	1.19	0.73	1.93	0.482	0.89	0.41	1.95	0.774
	18	As a categorical variable: HF (≤168 vs. >168)	0.62	0.24	1.61	0.330	1.28	0.61	2.66	0.517	0.62	0.19	1.96	0.412	0.46	0.06	3.35	0.445
	19	As a continuous (LnLF/HF)	0.98	0.62	1.56	0.932	1.08	0.79	1.48	0.617	0.70	0.40	1.22	0.207	1.72	0.67	4.43	0.261
	20	As a categorical variable: LF/HF (≤3.5 vs. >3.5)	0.95	0.39	2.31	0.901	0.91	0.50	1.67	0.762	1.44	0.45	4.63	0.543	0.79	0.15	4.09	0.779

^a^ Excluded 84 drivers who had CVD history before 2006; ^b^ Model 1: Adjusted for age at first employment (≥45 vs. <45 years), body mass index (>30 vs. ≤30), education, drinking, smoking, exercise, time since first employment (years), and shift work; ^c^ Model 2: As Model 1 with additional adjustments for systolic blood pressure, LnCHOL, LnTG, LnHDL, and LnAC; ^d^ Each independent variable (1–20) was separately included in the models.

## Data Availability

The data are not available, because we did not inform the participants of the data transparency nor declare the possibility on the institutional review board.

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
