# Peer review of "Identification of a High-Risk Group of New-Onset Cardiovascular Disease in Occupational Drivers by Analyzing Heart Rate Variability"

_ijerph, 2021, doi:10.3390/ijerph182111486_

Round 1

Reviewer 1 Report

Dear Authors,

With your study, you have examined an occupational group whose working conditions are still insufficiently studied. In particular, the cardiovascular risk does not seem to be low in this group. For my part, I have a few comments:
First of all, the study has some methodological weaknesses:
- Please add a limitation section at the end of your manuscript.
- Not all abbreviations are written out at the first mention in the text. Also, not all abbreviations are listed in the list of abbreviations. 
- No consistent reference style is used.
- Please mention explicitly in the methods section which cardiovascular diseases or risk factors were considered in the study.
- When did you perform the studies? At the beginning of the study and what follow-up period? Was it the same follow-up period for all subjects? How many study time points were there. Please also clarify this in Figure 1.
- Please support your statement made in lines 51-53 with literature references.
- Why did you study total cholesterol and HDL-C only? Why not also LDL-C?
- You talk about 8-year CVD risks in subsection 3.1. How was the risk determined? Did you use risk calculators?
- In Table 1, moderate exercise is mentioned as a parameter. Define this parameter. What do you mean by it?
- It would also have been interesting to investigate if there are differences regarding shift workers and non-shift workers? Could you add such a calculation? After all, 47% were irregular-shift working drivers.
- Lines 214 and 215: Please put your statement into perspective. They only did a 5 min HRV test. You could only make this statement if you did both methods (24 hr HRV-test and 5 min HRV-test) on the same cohort. Then a sufficient comparison would be possible .

Author Response

Thank you for your letter and for the reviewers’ comments concerning our manuscript. Those comments are all valuable and very helpful for revising and improving our paper, as well as the important guiding significance to our research. We have studied comments carefully and have made correction which we hope to meet with approval. Revised portion are marked in red in the paper. The main corrections in the paper and responds to the reviewer’s comments are as the attached file. 

Sincerely,

Wang, Ying-Chuan

Reviewer 2 Report

It is not clear how the authors have handled the artefact problems. Thus, we need to know explicitly how they handled arrhythmias and disturbances caused by movements and broken contact with electrodes. The most important one is how they handled arrhythmias that cause serious interpretation difficulties.

The authors reveal a remarkable lack of knowledge regarding the literature on work stress and cardiovascular disease. During later years several very large prospective studies have been published which show that job strain, lack of control, effort reward imbalance and shift/night work as well as excessively long work hours are associated with increased risk of developing ischemic heart disease. It is not defendable to write as the authors do:

however, we do not yet know whether work stress affects the development of CVD more than it contributes to reduced HRV.

I recommend the authors to do searches in pubmed on ischemic heart disease and each one of the following search words : job strain, job control, night work, shift work, effort reward imbalance. One cannot publish a study on heart rate variability and stress in bus drivers without this general view of the scientific situation. 

I also think the authors should quote Collins & Karasek who did a study of HF and low control at work. The original part of that study was that it was looking at everyday variations.

Author Response

(The authors gave the same response as above.)

Round 2

Reviewer 1 Report

An appropriate correction to the notes has been made. From my point of view, the manuscript can be published in this form.

Author Response

Dear reviewer,

Thank you so much for your comments and help!

BR,

Wang

Reviewer 2 Report

I still feel that the authors have not discussed the artefact problem sufficiently. They now mention that arrhythmias could be a problem in the interpretation  of the recordings. Other research groups spend a lot of time trying to sort out this problem. It is true that this study started before automated programs for artefact handling were widely available but there are even puritans in the field who say that computer identification of segments with aberrant patterns should be identified and interpreted manually. I do not require that but the authors should give more attention to this general problem in their discussion.

One of the more interesting observations in this study is that hypertensive disease is the cardiovascular disease that can be predicted when HRV is low. As a reader I wonder how many individuals there were in the different cardiovascular categories. The lack of significance for some of the other categories could be a power problem. Also, one starts wondering why there is such a clear relationship with hypertensive heart disease. Apart from numbers of subjects which could be a factor, does this reflect changes in baroreceptor sensitivity, with early HRV-disturbance being part of a blood pressure regulation problem? In addition, is it possible to say something clinically  about this hypertensive heart disease group? Do they have large hearts?

I have one more question: Since two factors of relevance for the general question about working conditions and heart disease among bus drivers have been recorded - shift/night work exposure and duration of employment - it would have been interesting to know something about the independent contribution of those factors to low HRV and high heart disease risk in this cohort. I am sure the authors must have done such analyses.

Author Response

Dear Reviewer,

Thank you for your letter and for the reviewers’ comments concerning our manuscript. Those comments are all valuable and very helpful for revising and improving our paper, as well as the important guiding significance to our research. We have studied comments carefully and have made correction which we hope to meet with approval. Revised portion are marked in red in the paper. The main corrections in the paper and responds to the reviewer’s comments are as the attachment.